# Removal of Stair-Step Effects in Binder Jetting Additive Manufacturing Using Grayscale and Dithering-Based Droplet Distribution

**DOI:** 10.3390/ma15113798

**Published:** 2022-05-26

**Authors:** Christoph Hartmann, Lucas van den Bosch, Johannes Spiegel, Dominik Rumschöttel, Daniel Günther

**Affiliations:** Fraunhofer Institute for Casting, Composite and Processing Technology IGCV, Lichtenbergstr. 15, 85748 Garching, Germany; lucas.bosch@tum.de (L.v.d.B.); j.spiegel@hawe.de (J.S.); dominik.rumschoettel@igcv.fraunhofer.de (D.R.); daniel.guenther@igcv.fraunhofer.de (D.G.)

**Keywords:** additive manufacturing, binder jetting, 3D printing, sand printing, casting, stair-steps, surface quality, grayscale, dithering, finishing

## Abstract

Binder jetting is a layer-based additive manufacturing process for three-dimensional parts in which a print head selectively deposits binder onto a thin layer of powder. After the deposition of the binder, a new layer of powder is applied. This process repeats to create three-dimensional parts. The binder jetting principle can be adapted to many different materials. Its advantages are the high productivity and the high degree of freedom of design without the need for support structures. In this work, the combination of binder jetting and casting is utilized to fabricate metal parts. However, the achieved properties of binder jetting parts limit the potential of this technology, specifically regarding surface quality. The most apparent surface phenomenon is the so-called stair-step effect. It is considered an inherent feature of the process and only treatable by post-processing. This paper presents a method to remove the stair-step effect entirely in a binder jetting process. The result is achieved by controlling the binder saturation of the individual voxel volumes by either over or underfilling them. The saturation is controlled by droplet size variation as well as dithering, creating a controlled migration of the binder between powder particles. This work applies the approach to silica sand particle material with an organic binder for casting molds and cores. The results prove the effectiveness of this approach and outline a field of research not identified previously.

## 1. Introduction

Sachs initially developed the binder jetting process for printing molds for metal casting [1]. Here, a layer of particle material (powder) is applied on a build platform, a print head deposits a bonding liquid, the build platform lowers, and the process repeats until the part is finished. Print heads usually contain an array of nozzles for rapid patterning. Therefore, the process is easily scalable by increasing the number of printer nozzles. Because of this, binder jetting is widely considered to have high deposition rates at a relatively low cost [2,3].

### 1.1. Binder Jetting for the Production of Sand Cores and Molds

Binder jetting enables the use of additively manufactured cores and molds for casting complex metal geometries. Here, a temperature-resistant powder material is combined with a polymer binder. After the process, the unbound powder is removed from the cavities using compressed air. After the assembly of the parts, molten metal is cast into the mold, and the casting is created [3,4].

Combined with conventional casting, binder jetting offers flexible tooling compared to machined patterns for molds and cores. Additionally, the geometric freedom offered by the process enables the casting of complex metal structures that are not possible to fabricate in traditional casting [4,5].

### 1.2. Stair-Step Effect in Additive Manufacturing

The stair-step effect is an implicit phenomenon of the layer-by-layer approach taken in Additive Manufacturing (AM). As previously described, the application of additional powder occurs as a sequential series of flat layers. These layers create a part with a stepped surface, shown in Figure 1 [2,6,7,8].

Triantaphyllou [6] describes the primary factors affecting the stepping of the surface:Layer thickness;Local surface slope;Material;Particle size range and distribution.

Furthermore, Triantaphyllou [6] shows the difference in surface quality between upskin and downskin surfaces, referring to the upper and lower surface considering the build direction.

Efforts have been undertaken to predict surface quality by modeling the stair-steps [9,10]. Kumbhar describes the stepped surface as a “major problem” [8] and documents the state of the art of post-processing techniques to improve the surface. Depending on the initial AM process, several post processes can be applied: abrasion finishing for stereolithography parts [11], vibratory grinding, partial surface melting, laser polishing, robotic finishing for SLS parts [12,13,14,15] and many more.

While there is a significant amount of literature on the post-processing of other AM processes, the post-processing of binder jetting parts usually refers to the curing cycle, depowdering, burnout steps and others [16]. Only a few publications were found regarding the post-processing of binder jetting parts to improve surface quality. For example, Godbey [17] describes the application of an additional surface layer consisting of finer particles to improve surface finish.

### 1.3. Literature on In-Situ Smoothing of Stair-Step Effects via Process Parameters

Many researchers assumed post-processing steps to be undesirable and focused on reducing the stair-step effect via in-situ process changes. As described by Triantaphyllou [6], reducing the layer height is an obvious way to improve the surface quality. However, the minimum layer thickness is closely tied to the largest occurring powder particles. Even though it might increase the overall cost of the process, it is a suitable way to reduce the stair-step effect. The following paragraph outlines alternative measures.

Adaptive slicing and non-planar layers improve AM processes using determined toolpaths such as fused deposition modeling [18,19]. Most literature on the smoothing of stair-steps focuses on stereolithography. Johnson [20] describes light intensity control projected off a digital micro-mirror array to improve surface texture and sharpness. Arif [21] utilizes a technique called slant beam rotation, in which an inclined beam of ultraviolet light can produce slant edges of individual layers and change the build angle. Park introduced an approach in which he utilizes a dithering approach to achieve adjustable beam intensity [22,23]. This adjustable intensity allows the cure depth of the resin to be controlled within a layer. The effectiveness of this approach was demonstrated in a micro stereolithography process.

To the best of the author’s knowledge, no published research covers the in-situ smoothing of stair-steps during the binder jetting process.

### 1.4. Approach and Theory

This work describes a novel method for smoothing stair-step surfaces of binder jetting parts. The literature describes fluid infiltration in porous media due to capillary pressure. For example, Barui investigated the ink infiltration of powdered materials similar to a binder jetting process [24]. It is concluded that the “infiltration proceeds in a highly non-uniform manner with dramatic differences in fluid penetration rate” [24] and that the Washburn/Denesuk [25] model can be applied successfully.

To vary the binder content (and therefore alter the effect of fluid migration in the outer layers of the part), we used a combined approach of using a grayscale print head and bitmap dithering. With the grayscale print head, different droplet sizes can be produced depending on the individual grayscale value of the pixel in the corresponding bitmap (Figure 2a,b). The dithered high-resolution bitmap allows for smoothly distributed binder content across the layer (Figure 2c,d) to produce local binder saturations not covered using the available grayscale values and drop sizes. In this way, we can investigate the suitability of varying binder content to address surface smoothing. Since the gravity-assisted pressure is negligible compared to the capillary pressure, the effect should show a similar effect both on upskin and downskin surfaces [26], barring effects caused by the powder recoating.

In the first step, we evaluated the printed specimens. Secondly, we adapted printing parameters to a printed cast mold to assess the suitability for casting. The resulting casting part can then be measured, and the effect on the smoothing of the stair-step effect can be evaluated.

## 2. Materials and Methods

### 2.1. Materials

The powder material used in this study is a commercially available silica sand with >99% purity (D_10_ = 104 µm, D_50_ = 153 µm and D_90_ = 223 µm). This powder is premixed with 0.3 wt. % of liquid commercially available acid containing mainly p-toluenesulfonic acid. The commercially available binder deposited by the print head consists mainly of furfuryl alcohol to be catalyzed by the acid to form a furan resin. All materials are commonly used in the industry. We set the time for curing to 24 h at room temperature.

A conventional commercial aluminum alloy EN AC-42100 is chosen for the casting part.

### 2.2. Printing Setup

Figure 3 shows the schematic binder jetting process as implemented in this work. The process itself is fully automated. First, the build platform lowers. Then, a layer of powder is placed on the build platform by the powder container moving to the right (a). Next, the rotating roller flattens the powder, and subsequently, the print head deposits its droplets fed from the binder tank (b). The steps repeat until the part has reached the desired build height.

Figure 4 shows the experimental test setup. The moving carrier unit holds the powder container, the roller, print head and binder tank and moves as a single unit on a linear axis. The structural design follows a topology-optimized approach to increase stiffness while reducing its inertia along the travel direction.

All elements can be adjusted in height and tilt. On the left, the roller cleaner can remove excess powder from the roller. The print head cleaning station sprays a cleaning spray onto the print head, which afterward dries using a compressed air stream coming out of a flat nozzle. The ultrasonic bath can extensively clean the printhead if necessary.

Stepper motors drive all axes. The interface to access the print head, motors and valves is PC-based.

The utilized print head is a Dimatix StarFire SG1024 1C MC LA, which comprises 1024 nozzles distributed over eight separate rows, supporting 2-bit grayscale printing at 400 dpi resolution, with droplet deposition timing provided by a linear encoder. We set the print speed to 100 mm/s and held the distance between the print head and powder bed at 5 mm.

### 2.3. Waveform Design and Bitmap Preparation

For simple geometries, the bitmaps for each layer are manually created using GIMP 2.10 (www.gimp.org). For more complex geometries, parts are designed in Inventor 2022 and Netfabb Premium 2022 (Autodesk, Inc., San Rafael, CA, USA), sliced and translated into bitmaps. GIMP is used to edit the black-and-white bitmaps to include grayscale and dithering patterns.

Four different levels of droplet sizes are used, with level 0 corresponding to a white pixel producing no droplet output. A rising level number indicates a larger resulting droplet. Figure 5 shows the waveform used to create differently sized droplets.

We printed 1024 × 1024 pixel images onto a dry particle material placed on a scale (Kern PCB 250-3, Kern & Sohn GmbH, Balingen, Germany) to infer the droplet group weight and measured them individually, repeating this at least 21 times in the same setup. The measured values were not corrected for non-functioning nozzles, which only amounted to a few, and we observed fluid evaporation to be negligible.

A level 1 droplet output per nozzle is calculated to have a mass of 22 ng based on the measurements. A second pulse quickly follows a short first pulse to reduce any additional droplet ejection, resulting in a very small droplet mass. Level 2 utilizes a standard single pulse, creating a droplet group weighing 47 ng. A level 3 pulse resulting in a total droplet group mass of 142 ng consists of two individual pulses producing a group of three droplets closely followed by each other.

With the combination of dithering patterns and the large range of droplet group weights, we created a wide parameter window to finely adjust the binder concentration in each area of each layer. The grayscale setup allows printing with binder mass ratios from 1% to 6%. The dithering patterns allow any intermediate concentration between 0% and 6% to be printed. We use the combination of both in this work.

Figure 6 shows the bitmaps printed for the test plates (shown in Section 2.5, Figure 7) with smoothened surfaces to explain the approach. On the left (a), the upskin-series of bitmaps is shown. Starting with layer x, the corresponding bitmap is printed. The smoothing ends with x + 3. Similarly, the first bitmap for the downskin smoothing starts with y and ends with y + 3.

As preliminary experiments showed, one stair-step can be smoothed very effectively if the two layers containing the step are modified by grayscale or dithering. Each step is treated with the adopted range grayscale distribution.

### 2.4. Finishing

We removed the printed parts from the loose sand after 24 h. After a rough cleaning using brushes and compressed air, we brushed the surfaces as a final treatment to remove any sand particles with only intermittent or weak adhesion to the part.

### 2.5. Printed Parts

In this work, we utilize three different geometries. All parts are printed with a nominal layer height of 0.35 mm.

#### 2.5.1. Cubic Samples

To understand the general relation between binder concentration and geometric accuracy, cubes of 20.07 mm × 15.05 mm × 17.85 mm are printed. To ensure that the measurement of the z-accuracy only evaluates upskin deviations, the respective upskin samples are printed directly onto the build plate. That means the first layer of the print consists of a singular sand layer directly on the metal base plate to form a compacted, smooth surface devoid of any adhering extra grains of sand. Downskin samples are printed in such a manner that any loose powder does not cover the parts. This print strategy is adopted to ensure that no particles are adhering to the upskin surface of the cube. The average layer time was measured to be 14 s and kept comparable for all specimens.

These cubes are printed in two variations of print strategies. In one, only the surface layer fill is printed with the set binder content (and the inner volume with 2%) (here defined as surface layer fill), and in the other, the complete volume is filled with the set binder content (defined as complete layer fill).

#### 2.5.2. Test Plates

To transfer the fundamental knowledge from the cubes to a smoothing process strategy and test different approaches, a test plate of 125.4 mm × 65 mm × 14 mm was designed. Figure 7 shows the schematic geometry in the upskin-view. It represents a waved geometry with small slope angles referenced to the xy-plane.

#### 2.5.3. Casting Mold

We developed a mold design to evaluate the impact of the strategy on the whole process chain of a cast part. It consists of three parts referenced using cone shapes on opposite sides—the assembly measures 280 mm × 65 mm × 47 mm. We used the smoothing parameter setting yielding the best results. Figure 8 shows the schematic casting mold.

### 2.6. Casting

We poured the casting part at 720 °C. The cuboid-shaped casting measures 200 mm × 45 mm × 8 mm and is fed via four gates and vented via four air pipes. Figure 9 shows the part’s concept.

The upper side of the casting is directly opposite the downskin side of the sand mold, while the lower side faces the upskin side of the sand mold. There are six relevant areas on the downside and the upside of the casting with the aim of having the same geometric features. We used the smoothening parameter set to create two ramps, ramp 1 and ramp 2, which face different directions. Each ramp surpasses two layers, reaching a total 0.7 mm nominal height difference between the three layers. Therefore, they represent the smoothened equivalent of steps 1 and steps 2, in which we used conventional printing techniques to bridge the height difference. In the middle of the part, there are two sinusoidal surface waves. The waves are phase-shifted and designed to reach a total of 0.35 mm height difference between minimum and maximum. As described above, we designed the same features on the downside of the casting. The gates and the air pipes are placed in between the relevant areas.

### 2.7. Examination Methods

#### 2.7.1. Dimensional Accuracy of Cubic Samples

We measured the height of the finished cubes with a TESA Digico 610 MI digital dial gauge (Hexagon AB, Stockholm, Sweden) imparting approximately 1 N of vertical force with a flat steel inset of 4.8 mm diameter to assess the dimensional accuracy of the cubic samples. Z-dimensions are measured with a total set of 10 measurements evenly distributed on the part surface. We used the loss-on-ignition method to determine the residual binder content of each sample. In this method, the cube is heated to 500 °C for one hour. The difference in weight closely correlates with the amount of binder deposited into the part. Combined with the known amount of activator, we could determine the binder concentration and apply the method to every sample.

#### 2.7.2. Test Plates for Stair-Step-Smoothing

The test plates are visually inspected and documented with a commercial DSLR camera. Using point source illumination hitting the surface at a shallow angle, over or under-compensation of stair-steps is well visible. This publication focuses on the best-rated parameter set.

#### 2.7.3. Dimensional Accuracy of Casting Part

We used a GOM ATOS Compact Scan stereo camera pattern projecting 3D scanner with a measuring volume of 170 mm × 130 mm × 80 mm to assess the dimensional accuracy of the casting. The part was fixed on a rotating two-axis table, measured from five directions per side. We sprayed the part with a thin layer of white acrylic paint to provide a bright matte surface for accurate scan results. The data were exported to the open-sourcesoftware CloudCompare (https://www.danielgm.net/cc/) and oriented via the iterative closest point algorithm to compare the measured data to a blank stl-cuboid. The distances were calculated using the signed distance field approach with this method.

The raw data that captured each sand particle’s shape were simplified using quadric error metrics [27] and afterward smoothened via laplacian mesh processing [28] to extract the profile curves. We then cut down the point cloud to the relevant areas, and the x-coordinate was collapsed, creating overlapping points with different z-coordinates.

## 3. Results

### 3.1. The Relation between Dimensional Accuracy and Binder Concentration

Understanding the influence of the binder concentration on the dimensional accuracy of the part is crucial for the success of this approach. We evaluated the cubic samples to develop a model for the underlying physical phenomenon. Here, two different sets of measurements were taken: upskin and downskin z-dimension difference between nominal height and actual height. Furthermore, we investigated the binder deposition strategy. One group of samples had the nominal binder content only set in the last layer, while the other group was built with a constant nominal binder over the whole part. Figure 10 shows the results of this investigation.

The first observation is that the difference between the surface layer fill and the complete fill is almost negligible. The surface fill creates a more significant geometric difference for low binder concentrations (below 1%) in the upskin than the complete fill. For all other binder contents for upskin and downskin, both print strategies cause a similar appearance. This paper does not investigate the underlying migration model; the investigations shown are phenomenological.

Noticeably, the complete fill causes significantly higher differences for very high binder contents. It should also be mentioned that these extreme geometric differences of over 5% do not create a smooth surface but rather a rough, broken and uneven plane, including sparsely distributed sand grains bound to the surface. However, it appears that a modification of the surface layer might be sufficient to achieve the desired result of smoothing the stair-step, which would have a maximum nominal height of 0.35 mm.

The peak at 6% of the binder is due to the described uneven and rough surface, making the measurement inaccurate. Nonetheless, the results can be interpreted as a design guide for bitmaps. These bitmaps can be color graded in accordance with the printing setup and the experimental data to compensate for the stair-step effect up to roughly 0.5 mm.

### 3.2. Three-Dimensionally-Printed Test Plates with Smoothened Stair-Steps

When condensing the results of Section 3.1 into a bitmap based on dithering and grayscale images, a guideline for printing different slopes can be developed. Figure 11 compares the upskin surfaces of a conventionally printed part (with stair-steps) on the left and the smoothened part on the right.

The difference can clearly be observed. While there is a distinct difference between the layers in the conventional process, smoothing can successfully be achieved with the described approach. Judging by the edges of the part, a consistent grading of the slope is possible. The discoloration on the part’s surface corresponds to the binder saturation in the respective area and is not due to surface geometry.

Comparing the bitmap design of the upskin to the downskin surface, a less pronounced approach is possible due to the lower binder contents required to achieve the desired height difference. Figure 12 compares the conventionally built part (left) and the smoothened surface (right) for the downskin surface.

Again, the effectiveness of smoothing the stair-steps is clearly observable. For the downskin, less binder can be used, and therefore the discoloration is minor. A few stair-step artifacts can be found in a steeper slope of the smoothened part, which most likely can be leveled out in another iteration of the parameter set.

### 3.3. Finishing

The finishing of the parts shown in Section 3.2 was further investigated with additional prints to determine the effect of the person tasked with finishing on the resulting surface. We used a different geometry here: a singular ramp with the newly developed parameter set. The small experiment has shown that persons familiar with the finishing protocol achieve the same satisfying result. However, persons not appropriately instructed are not able to finish the parts to a good result. The issue is caused by adhesions of loose sand, which optically appear very dark and are not easy to spot. Since the strength is significantly lower than that of the bounded sand, further brushing can remove them. Figure 13 shows the results of the experiment.

### 3.4. Casting

The question remains whether the method of grading the binder content is suitable to sustain the original intent of the printed parts—casting. Generally, increasing the binder content of sand molds and cores increases the chances of gas defects due to the burning of the binder. Arguably, the overall binder content increases negligibly since the modifications are only done in the surface layer, and an additional volume of sand is bound inside the stair-step. The following experiment is supposed to demonstrate this. We designed the previously shown cuboid part for that. The bitmaps were then edited based on the design guidelines to demonstrate the approach’s effectiveness on various geometries.

Figure 14 shows the GOM 3D scans from the cast parts. On the left, the upside of the casting is shown next to the color-graded image. In the right of the figure, the downside is shown.

This illustration intends to show the graded profile of the smoothened surface. It can be observed that the ramps are smooth and without distinct steps. The slight color changes, signaling a different height, might be due to measurement uncertainties due to the alignment of the digital model with the measured point cloud. However, the amount of detail in the measurement is sufficient as the scans resolve individual sand particles and grinding marks of the saw blade used to remove air pipes and gates.

We extracted profile curves for all six relevant areas for each upside and downside for further detail, as shown in Figure 15.

Noticeably, there is no tilt on the surface for the area of steps 1 on the upside or downside. The boundaries are defined for both sides. Nonetheless, slight overcompensation on the downside can be observed. Furthermore, the vertical edges are flattened out. This is probably due to a combination of sand mold finishing combined with sand surface properties and the casting process.

Similar to the area of steps 1, the profiles of steps 2 are smooth and defined with flattened edges.

Changing focus to the smoothened surfaces, it is noticeable that the results in both ramp 1 upside and downside display a line profile without stair-steps but with a slight deviation in the z-direction, signaling a tilt or perpendicular curvature in the surface.

Interestingly, the profiles of ramp 2 up and downsides are under-compensated for the upside and overcompensated for the downside. In addition, a tilt on the downside is noticeable. Since the bitmaps for both ramps 1 and 2 were identical but mirrored, this might be attributed to the cleaning of the sand mold.

The sinus 1 areas for the upside profile represents a smooth sinus curve within the aimed boundaries. The downside, however, is overcompensated and shows a slight tilt.

For the sinus 2 area, it is noticeable that, similar to sinus 1, the upside is smooth and defined, whereas the downside shows a slight overcompensation.

To put the highly magnified 3D scan profiles into context, almost none of the measured tilts and over or under-compensations are noticeable in visual inspection or handling.

## 4. Discussion

The first results show the influence of the binder content on the geometric parameters of a part. They indicate that we can use the binder content to control the size of a voxel. However, the relationship between voxel size and the binder content is not conclusively explained from the data of this experiment. We can control the size in the z-direction. Nevertheless, the size in the xy-direction is the way to control the binder content. Thus, non-linear behavior has to be expected. In addition, we can see that downskin is more sensitive to the binder content variation, necessitating different process parameters for upskin than downskin surfaces.

The application of assigning grayscale pixels and dithering of bitmaps seems to be an effective way to eliminate stair-step defects, as the demonstrator plates indicate. The approach to manipulating the two layers involved in creating the stair-step is crucial for success. We found this strategy by experiment and do not fully understand it yet. Here an immense potential for further improvements is hidden. At the moment, this demonstrator shows only designed stair-step features. Proof for industrial parts has to be provided.

The most obvious influence on the geometric variations might be the manual finishing. Finishing process automation is only beginning to be implemented in the industry for binder jetting processes. Since the parts require brushing, an influence of the manual handling on the geometric precision is likely, as shown by our experiment. Nevertheless, the result shows that the effect can be seen in the treatment of all operators. An automated and reproducible finishing has to be developed for more intense investigations.

The casted part shows in an effective way what level of smoothing can be reached. It can be assumed that the modified binder content does not affect the resulting part. Especially in areas with less binder, no defects such as voids can be found. In areas with a high binder content, no entrapped bubbles appeared in this experiment. In future investigations, realistic parts have to be cast to determine the effect on features such as edges, text letters or fine structures.

Furthermore, the GOM 3D scans reveal an apparent superior quality of the downskin surfaces (referring to the upside of the casting part). The difference between upskin and downskin (or similarly, downside and upside) can be explained with the binder necessary to reach the geometric target. The upskin requires more binder, making the process slightly more unstable. Furthermore, the binder’s migration, meaning the liquid’s movement after depositing, is severely obstructed in the upskin direction. With an average layer time of 14 s, it takes a significant time until the next layer of particle material is recoated, and the recoating itself induces a mechanical interference with the previously printed layer of sand. However, the newly added layer of sand is necessary for the upward migration of the binder. It can be assumed that a shorter layer time would reduce the required amount of binder for bridging the same geometric difference. With modern binder jetting machines having significantly shorter layer times, this problem is expected to be reduced.

If the industrial application requires further improved geometric accuracy, another iteration of process development has the potential to increase it further. In addition, parts can be oriented freely within the build area in binder jetting: turning the part around will interchange all upskin and downskin surfaces.

The maximum bridgeable dimensional difference sets the current limitation of this approach. Figure 10 (right) shows a steep increase in the downskin dimensional difference in the z-direction. Here, the challenge is the surface quality resulting at high binder contents; i.e., more than 5%. Depositing this large amount of binder causes broken and flaky powder adhesions instead of smooth, firm surfaces. We observed these surfaces to be not accurately repeatable and assessed them as unsuitable for casting applications due to the high chance of abrasion during the process. Future works should focus on building a model to understand the binder migration and push these current boundaries.

## 5. Conclusions

We conducted an experimental study of the effectiveness of grayscaling and dithering on the smoothing of stair-steps. For this study, a silica sand particle material combined with an organic furan binder was used. A layer height of 0.35 mm was set as the baseline for the smoothing process. As a first step, we analyzed the dependence of the geometric accuracy on the binder content. Interestingly, a variation of binder content in the surface layer could achieve an almost identical geometric result compared to a complete change of binder concentration in the overall part for the build direction.

We could use the gained understanding of the impact of the binder concentration to print demonstration parts, in which the stair-steps were almost entirely removed in their upskin and downskin surfaces.

Then, the process parameters were transferred to a newly designed casting mold, and castings were poured. An in-depth analysis of a part showed the complete removal of stair-steps, though revealing imperfections that were not noticeable by visual inspection.

This result means that the complete removal of the stair-step-effect in binder jetting parts and castings made from binder jetting molds (and cores) is achievable. It becomes clear that the binder’s migration depends on the binder content of the individual voxel and can be controlled using controlled droplet masses by way of grayscaling and dithering.

This understanding results in a new perspective on the binder jetting process. It is possible to print structures in-between the set grid of xy-resolution and layer height, maintaining the same process regime, smoothening surfaces and creating new opportunities for further research.

## 6. Outlook

A critical generalization of this study’s approach should be discussed further: until today, binder jetting has been considered a process in which geometry is simply the sum of its voxels. The voxel size is determined by the bitmap’s resolution and the layer height. Two-dimensional geometric details are printed in each layer. Now, a binder content-sized feature exchanges the fixed voxel dimensions, allowing unevenly sized voxels in every layer. This realization may result in a new field of research in binder jetting.

There are many possible applications when understanding and possibly controlling this effect. For example, a process speed-up could be achieved using thicker base layers. Our approach could achieve this change with comparable or even better surface quality. Another way to use this strategy could be to harden selected volumes of the parts to achieve a better overall accuracy after the finishing procedure.

In the future, newly developed binders should consider the new requirements this research will define: controlling the migration of the binder and adjusting the binder’s properties accordingly. Significantly larger layer heights now appear feasible, which would otherwise be impossible to manufacture today.

Future work should address the development of a migration model for the binder suitable to be used in simulations. This work could then be expanded to new chemicals, base powders such as metal powders and new process approaches for productivity.

## Figures and Tables

**Figure 1 materials-15-03798-f001:**
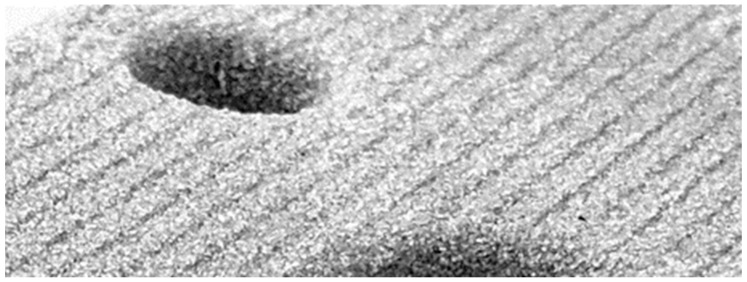
Stair-step effect in a sand core produced via binder jetting.

**Figure 2 materials-15-03798-f002:**
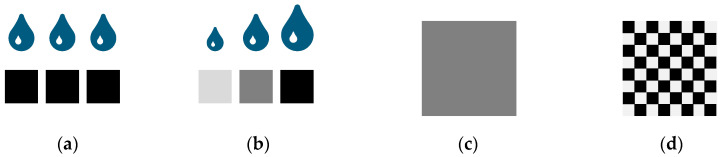
(**a**) Conventional droplet to pixel mapping with equal droplet sizes. (**b**) Grayscale printing with custom droplet size variation mapped to different shades of gray. (**c**) Grayscale image producing a single intermediate droplet size. (**d**) Dithered grayscale image containing only black and white pixels to simulate an intermediate droplet size not possible with the printhead or not covered using grayscale approach.

**Figure 3 materials-15-03798-f003:**
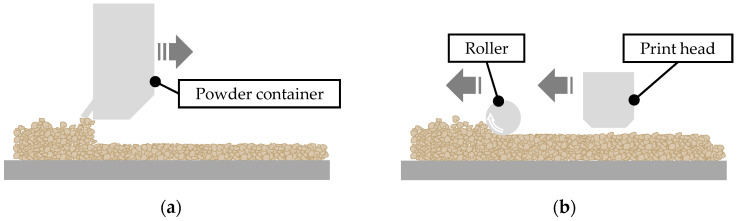
Schematic illustration of the recoating (**a**) and printing process (**b**).

**Figure 4 materials-15-03798-f004:**
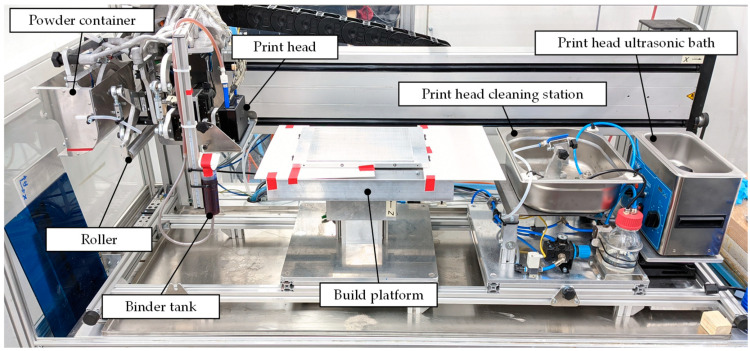
Binder jetting test setup used for the experiments.

**Figure 5 materials-15-03798-f005:**
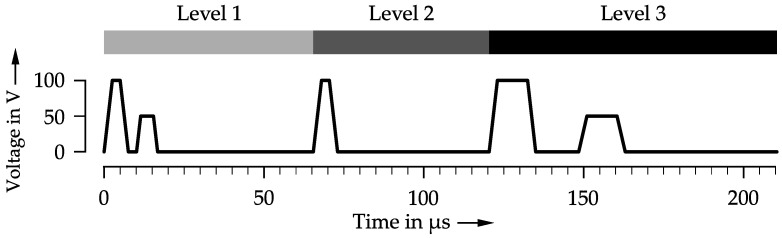
Piezo crystal control waveform to create three different droplet sizes for the Dimatix StarFire print head. Level 1 (light gray), level 2 (dark gray), level 3 (black).

**Figure 6 materials-15-03798-f006:**
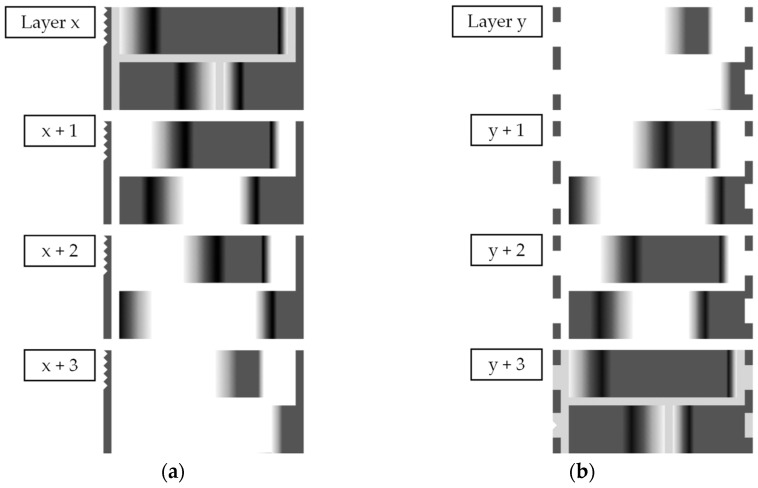
Grayscaled and dithered bitmaps printed for the parts in this work. (**a**) Upskin-series of bitmaps. (**b**) Downskin-series of bitmaps.

**Figure 7 materials-15-03798-f007:**
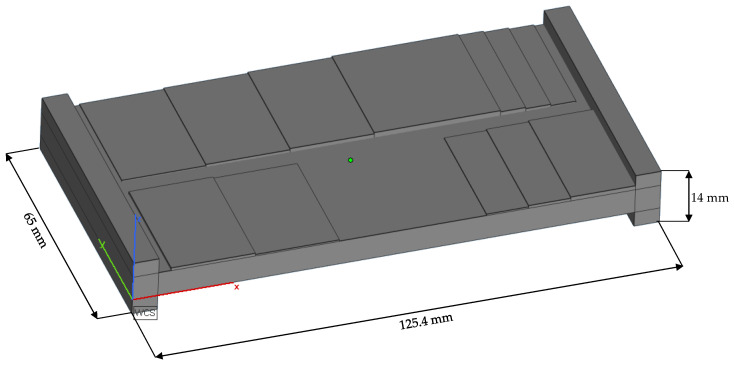
Schematic geometry of the test plates. The side shown is the upskin.

**Figure 8 materials-15-03798-f008:**
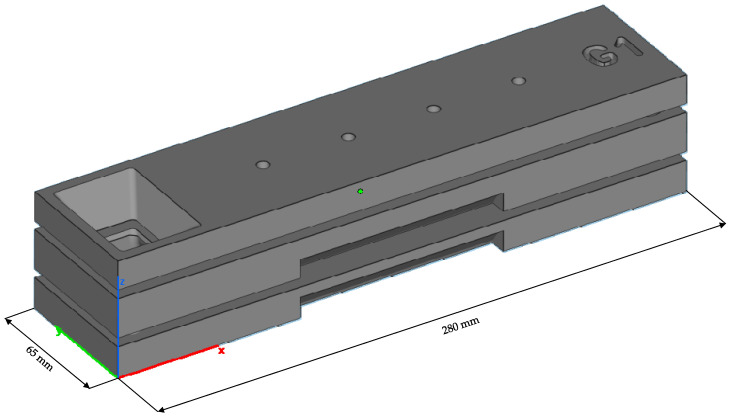
Schematic geometry of the three-part casting mold. The pouring basin is the hole on the left.

**Figure 9 materials-15-03798-f009:**
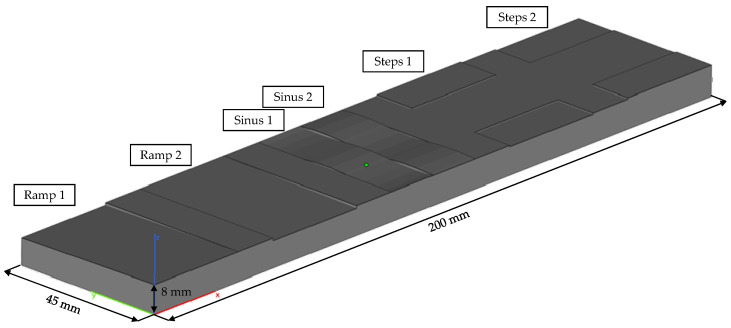
Schematic illustration of the concept casting part, upside view.

**Figure 10 materials-15-03798-f010:**
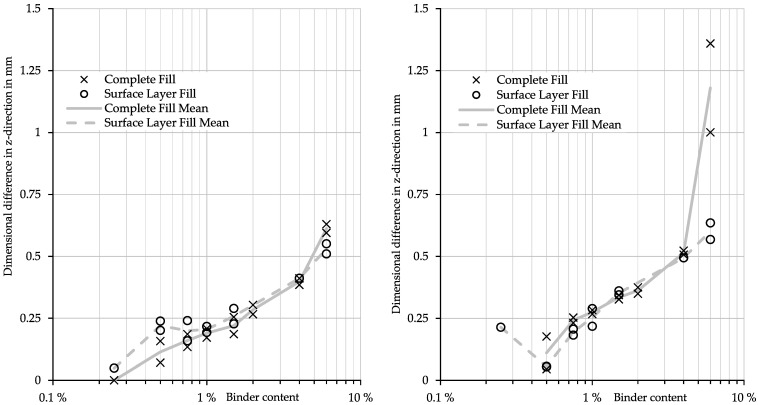
Influence of the binder content on the geometric difference between the nominal and actual dimension in the z-direction (**left:** upskin, **right:** downskin).

**Figure 11 materials-15-03798-f011:**
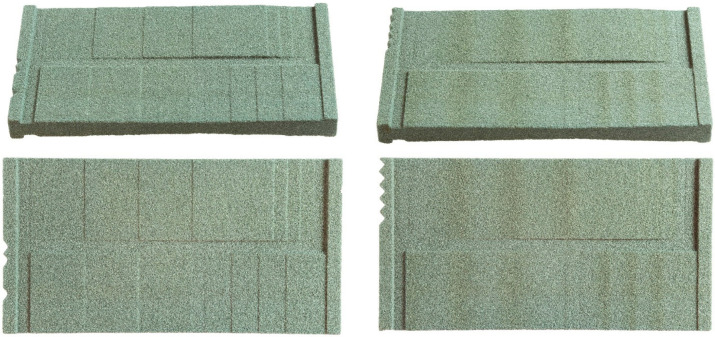
Two upskin perspectives on a conventionally printed part (**left**) and a smoothened part (**right**).

**Figure 12 materials-15-03798-f012:**
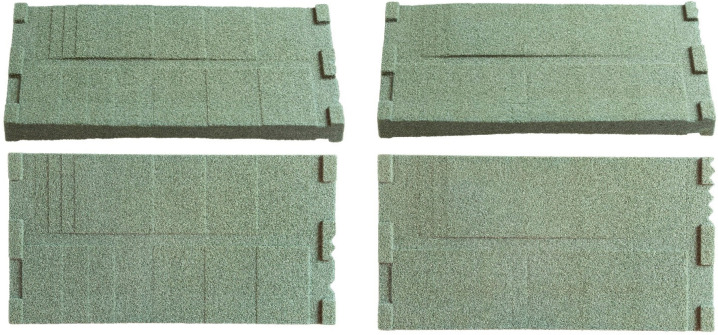
Two downskin perspectives on a conventionally printed part (**left**) and a smoothened part (**right**).

**Figure 13 materials-15-03798-f013:**
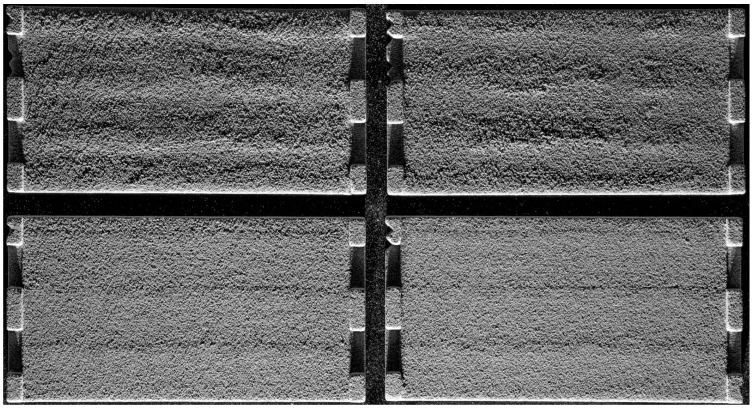
Four identical downskin smoothing parts finished by four different people. The upper two samples still contain a surface layer of loosely adhering sand, while the lower two are finished correctly with a higher surface strength remaining.

**Figure 14 materials-15-03798-f014:**
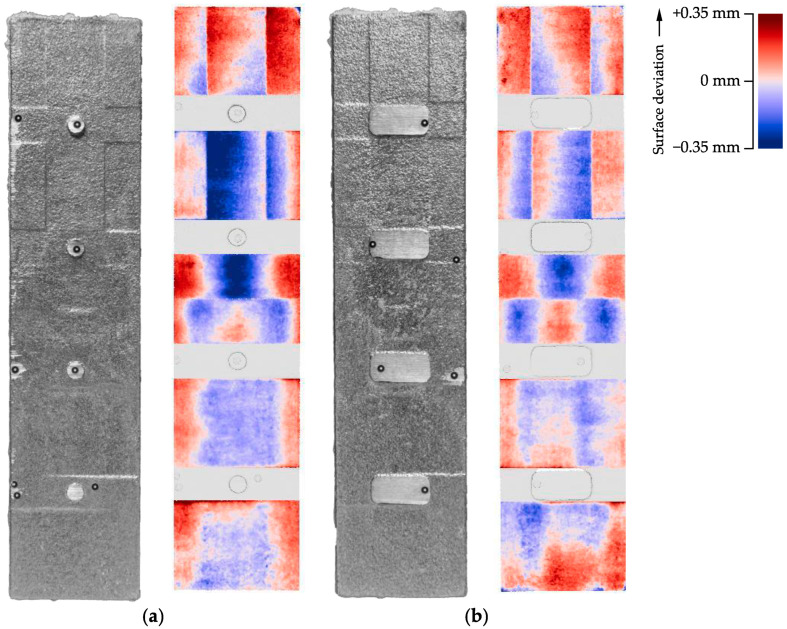
(**a**) Upside of cast part; GOM measurement of the upside; (**b**) Downside of cast part; GOM measurement of the downside.

**Figure 15 materials-15-03798-f015:**
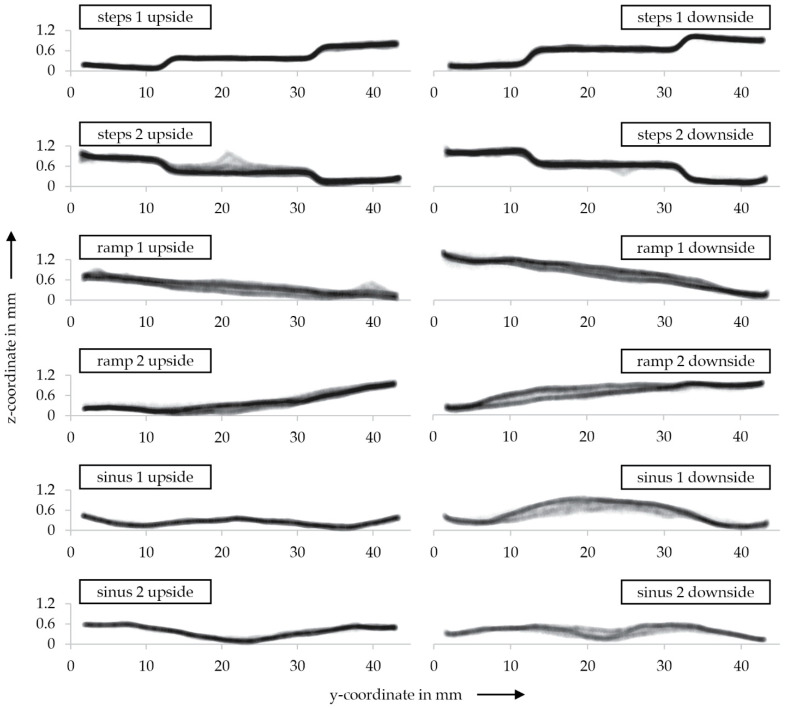
Profile comparison of both upside (**left**) and downside (**right**) for steps 1, steps 2, ramp 1, ramp 2, sinus 1 and sinus 2.

## Data Availability

The data presented are available on request from the corresponding author. The data are not publicly available since they are part of an ongoing study.

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
