# Peer review of "Removal of Stair-Step Effects in Binder Jetting Additive Manufacturing Using Grayscale and Dithering-Based Droplet Distribution"

_materials, 2022, doi:10.3390/ma15113798_

Round 1

Reviewer 1 Report

This manuscript reported a method to remove the stair-step effect in the binder jetting process. The method is applied and proved in the silica sand particle materials with an organic binder for casting molds and cores. In reviewing the manuscript, the following comments are made.

  1. Figure 7 is not needed in the article, because this cubic does not have a specific shape, dimension or orientation that need to be stated.
  2. In section 2.7.2, the manuscript shows or “focus on the best-obtained result”. Generally, to make the results reproducible, articles will prefer to present the results with average performance.
  3. It is recommended that integrate the Figure 16 to Figure 21 to make them better readability.

Author Response

Before we address your feedback individually, thank you for critically reviewing our work. Your comments were carefully considered and implemented in our manuscript. The changes in the manuscript are highlighted in red.

Point 1: English language and style are fine/minor spell check required.

Response 1: The manuscript underwent an extensive language check. We corrected several grammatical mistakes and tried to make it easier to read by using more active voice.

Point 2: Figure 7 is not needed in the article, because this cubic does not have a specific shape, dimension or orientation that need to be stated.

Response 2: Thank you for pointing this out. With the motivation to standardize the structure, this unnecessary figure was inserted. It is removed now.

Point 3: In section 2.7.2, the manuscript shows or “focus on the best-obtained result”. Generally, to make the results reproducible, articles will prefer to present the results with average performance.

Response 3: You are very right for pointing this out. Though here, we falsely used this phrase. All results shown in the manuscript were repeated several times with good accuracy. In the stated section, it should refer to the best parameter set of the parametric investigation – the parameter which fit our binder and printer the best.

Point 4: It is recommended that integrate the Figure 16 to Figure 21 to make them better readability.

Response 4: Thank you for this idea. We created a summarized figure where the profiles can more easily be compared.

Reviewer 2 Report

This paper has a clear objective and high engineering reference value. It presents a method to remove the stair-step effect entirely in a binder jetting process. This is achieved by controlling the binder saturation of the individual voxel volumes by either over or underfilling them.

The aim of this study is to provide a method to remove the stair-step effect entirely in a binder jetting process. This method is applied to silica sand particle material with an organic binder for casting molds and cores. The results prove the effectiveness of this approach and outline a field of research not identified previously. The authors need to address the following issues/comments for publications by the journal:

1.               Is this method only suitable for binder jetting additive manufacturing? As we all know, As defined in the ISO/ ASTM 52900:2015 standard, AM encompasses seven distinctly different process categorizations and each of these seven AM categories is differentiated by the form of feedstock and/ or the binding process used to join the material. Does this paper have any reference value for other additive manufacturing processes?

2.               Page 5, Fig.5: Pay attention to the sharpness of the image, especially the gray font. Fig. 16-21, same problem in gray font.

3.               Page 10, Fig.11: It is recommended that the ordinate of the two figures be the same. (The negative half of the ordinate seems meaningless.)

4.               In the DISCUSSION or CONCLUSION section, please add any limitations to the method described in this article.

Author Response

Before addressing your feedback points individually, thank you for taking time to critically review our work. We appreciate your feedback and implemented your suggestions in our manuscript. These changes in the manuscript are highlighted in red.

Point 1: Extensive editing of English language and style required

Response 1: Thank you for pointing this out. Though we did many rounds of language crosscheck, there were several occasions on which we could fix grammatical errors and word repetitions. Furthermore, we tried to use more active voice to make it easier for the reader to understand. We hope that these changes will change your mind on the quality of our use of English language and style.

Point 2: Is this method only suitable for binder jetting additive manufacturing? As we all know, As defined in the ISO/ ASTM 52900:2015 standard, AM encompasses seven distinctly different process categorizations and each of these seven AM categories is differentiated by the form of feedstock and/ or the binding process used to join the material. Does this paper have any reference value for other additive manufacturing processes?

Response 2: You are correct. Wrongly, we were tempted to use this standard phrase as an introduction to our manuscript. Since we are solely speaking about binder jetting, there is no need for any unwanted implication on other AM processes. The sentence is removed.

Point 3: Page 5, Fig.5: Pay attention to the sharpness of the image, especially the gray font. Fig. 16-21, same problem in gray font.

Response 3: Thank you for spotting this. The grey font was due to an export error and we did not see this. As the other reviewer suggested, we moved the Fig. 16-21 into one merged figure for easier reference and comparison. Here we paid attention to the black font. Also, additionally to Figure 5, we changed many images for vector graphics (or if not possible because too large for 300dpi images). Especially with the bitmap overview it is worth having higher quality figures.

Point 4: Page 10, Fig.11: It is recommended that the ordinate of the two figures be the same. (The negative half of the ordinate seems meaningless.)

Response 4: This is a good suggestion. We changed the ordinate to be covering solely the positive values and changed the height of the graph for better comparison.

Point 5: In the DISCUSSION or CONCLUSION section, please add any limitations to the method described in this article.

Response 5: Thank you for letting us know your questions while reading this manuscript. We added a paragraph in which we refer to the results of higher dimensional changes (larger than 0.5 mm) and how they change the surface quality of the part. In our point of view, this is the current limit for this technique because certainly these adhesions will not remain stable during the casting process and therefore will cause defects.